# The High-Temperature Soft Ferromagnetic Molecular Materials Based on [W(CN)_6_(bpy)]^2−/−^ System

**DOI:** 10.3390/molecules27144525

**Published:** 2022-07-15

**Authors:** Janusz Szklarzewicz, Maciej Hodorowicz, Anna Jurowska, Stanisław Baran

**Affiliations:** 1Faculty of Chemistry, Jagiellonian University, 30-387 Kraków, Poland; hodorowm@chemia.uj.edu.pl (M.H.); jurowska@chemia.uj.edu.pl (A.J.); 2M. Smoluchowski Institute of Physics, Faculty of Physics, Astronomy and Applied Computer Science, Jagiellonian University, 30-001 Kraków, Poland; stanislaw.baran@uj.edu.pl

**Keywords:** cyanides, ferromagnetic, structure, tungsten

## Abstract

The synthesis of molecular materials with magnetic properties, in particular ferromagnetic properties, has been the subject of interest in coordination chemistry for decades. In the last three decades, research has accelerated, as it has emerged that creating bridging systems based on cyanido ligands is a good and relatively simple way to create complex polymer structures exhibiting magnetic properties. Based on many years of personal experience in the field of the synthesis of polycyanido systems, supported by comprehensive structural analysis, a simple method of transforming cyanido complexes into soft ferromagnetic materials with a Curie temperature (T_C_) higher than the thermal decomposition temperature, usually above 150 °C has been developed. Two soft ferromagnetic materials based on zinc and cadmium hexacyanido salts in the system with [W(CN)_6_(bpy)]^2−/−^ anions are presented. The crystal structures (X-ray single crystal as well as XRD) of the precursors and the properties of the ferromagnetic materials are discussed. Most importantly, a patented method of synthesizing this type of material, based on which we obtain more than 80 soft, high-temperature ferromagnetic compounds, which proves the wide spectrum of this method, is also presented.

## 1. Introduction

The magnetic properties observed in the coordination compounds of *d*- and *f*-electron metals are called molecular magnetism [1]. Since the second decade of the twentieth century, this issue has been very popular due to the search for new, multifunctional materials with the properties of magnets, characterized by the so-called “magnetic memory”. Therefore, many research groups around the world have started a race to discover molecular magnetic systems with ferromagnetic properties at the highest possible temperatures, at least at the temperature of liquid nitrogen. Research accelerated when it was discovered how readily cyanido ligands, especially in transition metal complexes, combined with metal cations to form inorganic multidimensional polymers.

The properties of molecular magnets (nanomagnets) can be observed in isolated molecules containing a single paramagnetic ion arranged in clusters, coupling several or even several dozen paramagnetic ions, and in coordination polymers called single-chain magnets. In molecular systems, molecules retain their characteristics, which makes it possible to design materials with specific physical properties. Coordination systems belonging to the class of the so-called molecular magnetics exhibit great structural diversity. By using a variety of metal ions and organic or inorganic bridging ligands in their synthesis, it is possible to create compounds from single-core to multi-core clusters. The nature and strength of magnetic interactions in such systems depend on the type of metal ions used, their mutual spatial arrangement and the presence of bridging ligands and their geometry. In our research, we showed that even simple cation–anion systems can be responsible for the formation of complex structures and that there is no direct correlation between the anionic charge or the anion–cation ratio and the structure formed [2,3].

The most frequently used precursors of molecular magnets are paramagnetic complexes of transition metals with cyanido ligands. These are Mo(V), W(V), Nb(IV) octacyanides, Fe(III), Co(III) hexacyanides and others, as well as more advanced mixed-ligand complexes, e.g., a [W(CN)_6_(bpy)]^−^ ion complex synthesized by one of the co-authors [4]. As cyanido complexes are generally anionic, cyanido bridges are formed to the cations. To increase the number of unpaired electrons in the synthesis of such systems, the metals of the *d* and *f* block are used in particular. By changing the charge of anions and cations, or rather the type of metal used, various structures can be obtained, from monomers to two-, three- and multi-core compounds forming 0D systems, chain (1D) polymers, two- (2D) and three-dimensional (3D) and supramolecular structures, thanks to the presence of intermolecular interactions, in particular, hydrogen bonds. To date, tens of thousands of articles have been published on the subject, and more than a thousand are still published each year [5,6,7,8,9,10].

The greatest challenge is to obtain not only simple paramagnets but, in particular, ferromagnetic complexes with high Curie temperature (T_C_). Despite many attempts, the T_C_ temperature is still well below room temperature and in most cases well below the liquid nitrogen temperature. Until our invention, there have been no synthetic procedures that could guarantee the ferromagnetism of the product.

In this article, we present a simple, repeatable method of synthesizing ferromagnetic materials, both in the solid phase and in the form of ordinary liquids, developed over two years ago [11,12,13]. All these ferromagnetic materials are stable well above room temperature, in most cases well above 100 °C, often up to the thermal decomposition temperature (release of ligands). So far, we synthesized over 80 complexes with the properties of soft ferromagnetics [13]. Today, we can produce ferromagnetic material from almost any paramagnetic cyanido complex of any transition metal. So far, the method has been proven successful in all the investigated cases, and it can probably also be applied to systems other than cyanido complexes. As an auxiliary material, we present the ferromagnetic behavior of solid (powder) ferromagnetic complex as a video file.

## 2. Results

### 2.1. Design and Characterization

Advanced methods of crystal engineering are the key to the synthesis of molecular magnets. Cyanido complexes are typical starting materials in the synthesis of modern molecular magnetic materials. A short contact between paramagnetic centers is required to observe the magnetic order. The Me-CN-Me’ distance (where Me is the *d*- or *f*-electron) depends only on the radius involved in the metal bridge, since the linear cyanido bridge does not allow for less than 5 Å of Me-Me contact. This is a serious problem in the synthesis of ferromagnetic materials, and the only possibility to reduce the distance between paramagnetic centers is to use metal atoms with a high coordination number and the formation of as many cyanido bridges as possible to create the Me–Me side-on interactions. Therefore, it is believed that complexes based on mixed ligands, including cyano-organic ones, are not good precursors of magnetic materials. We decided to take a different approach, so we wondered whether complexes containing cyanido and, e.g., bipyridyl ligands, are suitable for the synthesis of molecular ferromagnets. What will be the influence of ligands other than cyanido on the arrangement of the structure of the tested compounds? Will the presence of a bulky ligand containing the ring system cause the emergence of new intermolecular interactions that will force an unusual arrangement of cyanido bridges and thus cause paramagnetic centers to come closer together?

In the manuscript, to show the possibilities of our method, we used the worst starting materials, in terms of ferromagnetic properties, that we could find, i.e., we used diamagnetic Zn^2+^ and Cd^2+^ as cations and [W(CN)_6_(bpy)]^2−/−^ as anions. Thus, only one metallic center, in our case, W(V) with the *d*^1^ configuration, could be responsible for the ferromagnetic interactions. However, due to the large W-W distance [W-CN-(Zn/Cd)-NC-W distance] of more than 8 Å, in particular, 8.168 or 8.159 Å for Zn and Cd, respectively (see cif files), at first look, such interactions are excluded. We also used an anion complex, synthesized by one of us in 1988, containing mixed ligands of the anionic formula [W(CN)_6_(bpy)] with 2,2′-bipyridine as a ligand blocking the formation of a complex 3D system with cyanido bridge interactions [14]. This ligand should introduce steric hindrances to the surroundings of the metallic center of W(V), thus reducing the possibility of magnetic interactions. The use of cations with the *d*^10^ configuration also excludes the problem of high/low spin change in polymer products. Thus, we did our best to prevent the synthesis of ferromagnetic materials, following the typical thinking on molecular magnets.

We present here the structure of three salts with Zn^2+^ and Cd^2+^ cations and anions [W(CN)_6_(bpy)]^2−/−^ used in the synthesis of soft ferromagnetic material. We also present preliminary evidence of its high-temperature ferromagnetism and a possible explanation of ferromagnetic interactions. Our patented method requires several critical conditions to achieve ferromagnetic interactions. First of all, the most obvious is the use of paramagnetic centers. In the presented case, we used the anion [W(CN)_6_(bpy)]^−^ with the *d*^1^ configuration. This anion has a typical magnetic moment associated with one unpaired electron (approx. 1.73 μ_B_). The second condition is the use of a cyanido complex, as cyanido ligands serve as strong bridges between the cations and the anions that make up the polymer structures, which is typical in all papers dealing with molecular magnets. The third condition is the use of cations that coordinate with the nitrogen end of the cyanido ligands and form a polymeric, insoluble precipitate in a fast reaction (Zn^2+^ or Cd^2+^ cations, in our case; in the literature, typically, paramagnetic cations are used to observe metal–metal interactions in the Me-CN-Me chain). The following three conditions, we discovered, are the most important. These are: (1) the applications of the diamagnetic analog of the paramagnetic center (in the present case, it is an anion [W(CN)_6_(bpy)]^2−^); (2) the diamagnetic anion must also form an insoluble precipitate with the cation used, across the cyanido bridges; and (3) the product must have lower solubility than its paramagnetic counterpart. All these conditions are met by the two systems described here. The structures of diamagnetic and paramagnetic precursors are the most important.

### 2.2. The structures of Precursors

Due to requirements of low solubility of the formed complexes, after several months of growth under special conditions, we were able to isolate single crystals of three salts, which are precursors of two final products showing magnetic properties. These are for W(V):[Zn(bpy)Cl][W(CN)_6_(bpy)] (**1**) and [Cd(bpy)(H_2_O)(NO_3_)][W(CN)_6_(bpy)] (**2**) and one for W(IV), [Cd(bpy)(H_2_O)][W(CN)_6_(bpy)]·2H_2_O (**3**). Unfortunately, despite over a year of attempts to obtain crystals of the Zn analog from W(IV), it was not possible to obtain crystals of sizes suitable for diffraction measurements on a single crystal. The product can be precipitated as a very fine orange powder (**4**). The structures of both W(V) complexes are very similar and show 1D molecular chains, presented in Figure 1 for the Cd analog (extended data, as well as the structure of the Zn analog, can be found in Appendix A). The distance between the paramagnetic centers (8.168 Å for **1** and 8.159 Å for **2**) is long enough to prevent spin–spin interactions between tungsten atoms, and both salts show typical magnetic moments of ca. 1.73 μ_B_. The coordination environment around the cations adopts the geometry of a slightly distorted trigonal bipyramid (**1**) or a prism (**2**), while the complex anion in all the described compounds adopts the geometry of the dodecahedron. Importantly, the anions in structures **1** and **2** only use two cyanido ligands at positions 2 and 4 as cation-bridging ligands. In addition, half of the cation charge is neutralized by a cation-labile anion (Cl^−^ at **1** or NO_3_^−^ at **2**). The W-W distance is shorter than expected for W-CN-Cd-NC-W, since the N-Cd-N angle is 80.18°. One-dimensional W-Cd or W-Zn chains are separated by bpy molecules, as shown in Figure 1c. These chains are linked by the π–π stacking interactions, which results in a separation of adjacent W atoms by at least 9 Å (chains separated by bpy ligands in the complex W) or by more than 12.965 Å when the W atoms are separated from the Cd atoms by the bpy ligand (**2**).

The structure of the diamagnetic salt **3** of W(IV) with the d^2^ configuration shown in Figure 2 is much more complicated because three cyanido ligands of the [W(CN)_6_(bpy)]^2−^ anion are involved in the formation of cation-anionic bridges. The structure of the described compound is of the 2D type, with the layers separated by molecules of water of crystallization. The coordination environment of the Cd^2+^ cation adopts the geometry of an octahedron with a coordinated one-labile water molecule. The shortest W-W distance in **3** is 8.509 Å.

### 2.3. The Ferromagnetic Transformation

We present here the procedure of ferromagnetic materials’ synthesis based on zinc and cadmium salts—**5** with Zn^2+^ and **6** with Cd^2+^ cations. The most important fact is that the creation of the final product begins with the formation of an appropriate mixture of cyanido precursors, complexes of W(V) and W(IV) in the form of their PPh_4_^+^ salts, well soluble in a H_2_O-MeCN mixture. To this mixture, zinc or cadmium nitrates water solution is added. Due to much lower solubility of the W(IV) salts, they start to precipitate, at first forming diamagnetic zinc or cadmium salts of the [W(CN)_6_(bpy)]^2−^ ion {due to very low solubility of zinc salt, we obtained too small crystals for X-ray measurements, but the cadmium salt had a structure described earlier (complex **3**)}. As the salts precipitate, the W(IV) anion concentration decreases, and the W(V) cadmium or zinc salts start to co-precipitate, but as cations, in the earlier precipitated W(IV) salts crystals, remaining coordinatively unsaturated. The W(V) anion tends to coordinate with cations (zinc or cadmium) on the W(IV) substrate. This results in forcing the W(V) zinc or cadmium salts to mimic the structure of the W(IV) substrate, at least to some extent. This process is responsible for the closer W-W distances and formation of ferromagnetic interactions between W(V) centers in the material formed. With the restricted method that we discovered, we were able to increase the magnetic moment of this complex, without changes in the substrates used, from typical molar content in the product for a d^1^ system (from expected 1.45 μ_B_ for 80% of W(V)) to ca. 12.1 μ_B_ for **5** and 11.1 μ_B_ for **6** per tungsten atom, without change in the substrates used. In terms of magnetic properties, the increase in gram susceptibility was from 1.67·10^−6^ to 1.03·10^−4^ for **5** and from 1.67·10^−6^ to 8.69·10^−5^ for **6**. We were unable to determine the Curie temperature, as it was found to be higher than the complex decomposition temperature; still, the product retained its magnetic moment unchanged (within experimental error) up to 150 °C. In dc magnetic measurements, we did not exceed 300 K, as water of hydration was released (see TG curves in Appendix A). This could have contaminated the measuring chamber.

It must be stressed that the high magnetic moment depends strongly on the synthetic conditions, and small changes can result in a dramatic decrease in the magnetic moment of the samples. It is obvious that the small dimension of former crystals of the W(IV) precursor is crucial for its high surface area, and thus, the presence of numerous places for W(V) deposition. We found that ferromagnetic materials **5** and **6** are very sensitive to the solvent presence, and even acetone added to the ferromagnetic product decreases the magnetic moment dramatically, even up to 1.45 μ_B_ per tungsten atom, thus to a value expected for the paramagnetic sample. After solvent removal, the magnetic moment increases, and the ferromagnetic properties return to their original value.

Figure 3a,b presents the IR spectra, in the ν_CN_ range, of **1**–**4** precursors compared to ferromagnetic samples **5** and **6**. As in general, the ν_CN_ bands are very sensitive to symmetry, the oxidation state of metal and the types of bridges formed, bands in this region serve as a possible source of ferromagnetic material structure. As can be seen, the spectra of ferromagnetic salts are the sum of bands of precursors. This may indicate that ferromagnetic materials preserve the types of interactions and structures observed in the precursors. As the W(V) ν_CN_ bands are of very low intensity compared to the W(IV) ones, the bands of this last precursor are mainly observed.

We also present the powder diffractograms of the precursors and ferromagnetics **5** and **6** (Figure 3c,d). It can be seen that ferromagnetic materials **5** and **6** are crystalline (the background is caused by the apieson used to immobilize the samples). The substantial structural changes in the ferromagnetic materials compared to their precursors can be observed. W(V) salts are formed on W(IV) crystals, preserving, at the beginning of crystallization, the Me^2+^-Me^2+^ distances; thus, the substantial changes in powder diffractograms are expected. However, as W(V) is present in 4:1 excess over the W(IV) complex, gradually, the more distant from the crystal surface structure of W(V) salts of Me^2+^ observed in **1** or **2** begins to recover, as seen in Figure 3.

## 3. Discussion

The [W(CN)_6_(bpy)]^2−/−^ anion salts (both for Zn^2+^ and Cd^2+^) show typical magnetic properties of d^2^ (diamagnetic—complexes **1** and **4**) or d^1^ (paramagnetic—complexes **2** and **3**, with magnetic moment of 1.7 m_B_) systems, respectively. Surprisingly, when zinc or cadmium salts are precipitated from a mixture of 2- and 1- anions, the magnetic moment does not decrease and shows values lower than 1.7 m_B_ (due to the amount of d^1^ anions in the product being lower than 100%) but increases rapidly to over 10 per tungsten atom (complexes **5** and **6**). This indicates that ferromagnetic interactions between the W(V) centers arise. This high magnetic moment is found to be present up to the thermal sample decomposition temperature. Up to now, no one has observed such an interaction, and this is why we patented the method of synthesis of those systems. The initial squid measurements confirmed the ferromagnetic behavior (see Figure 4). The hysteresis is very narrow; this is characteristic for soft ferromagnetic materials. Similar behavior is also observed for sample **6** (with Cd). Those measurements require further deep interpretation and more extended measurements, which are in progress. We indicated in the manuscript that, probably, the structure of the less soluble W(IV) salts promotes the proper rearrangement of the W(V) salt deposited in the second step on the W(IV) template. Further intense studies of those fascinating but very challenging systems are in progress.

## 4. Materials and Methods

### 4.1. Chemicals and Materials

All solvents (except ethanol), agar-agar, ZnCl_2_·6H_2_O and Cd(NO_3_)_2_·4H_2_O were of analytical grade (Sigma-Aldrich, Madison, WI, USA) and used as received. Ethanol (97%) was from Polmos (Poland) of pharmaceutical grade and was used without further purification. (PPh_4_)_2_[W(CN)_6_(bpy)]·4H_2_O and (PPh_4_)[W(CN)_6_(bpy)] were synthesized as described earlier [4,14,15].

### 4.2. Synthesis

In general, two methods were used. Method (a) was used for single-crystal synthesis, while method (b) was used in the synthesis of a larger amount of the product (see also Figure 1).

*Method (a)* An amount of 1 g of agar-agar was dissolved in ca. 50 mL of water at 90 °C. A 2 M aqueous solution of zinc or cadmium salt (0.5 mL) was mixed with 4 mL of the prepared agar-agar solution. The mixture was heated up to ca. 80 °C and sonicated for 5 s. The transparent solution was then placed in a glass vial, forming a ca. 1 cm layer. The mixture was cooled in the refrigerator (at −20 °C) until the jelly hardened. Then, the agar-agar solution (forming a ca. 3 cm layer) was placed on the top of the first layer and, again, it was cooled until it hardened. Next, the concentrated solution of tetraphenylphosphonium salt of [W(CN)_6_(bpy)]^2−^ or [W(CN)_6_(bpy)]^−^ in MeCN-H_2_O (1:3, *v/v*) was mixed with the agar-agar solution (1:2, *v/v* ratio) and sonicated for 5 s. This mixture was used to form the last layer (of ca. 1 cm thickness) in a glass vial. Appendix A presents the photo of such prepared set. This was left in the dark for several months to grow. Single crystals were manually separated from the agar-agar.

*Method (b)* An amount of 0.1 g of tetraphenylphosphonium salt of [W(CN)_6_(bpy)]^2−^ or [W(CN)_6_(bpy)]^−^ was dissolved in MeCN (5 mL), and 15 mg of bpy was then added. Next, 2 mL of 1M aqueous solution of the respective salt of zinc or cadmium was added. The precipitate was filtered off, washed with water, ethanol, acetone and dried in the air. In the case of the W(V) salt, all operations were performed in a dark room.

#### 4.2.1. Synthesis of [Zn(bpy)Cl][W(CN)_6_(bpy)] (**1**)

(a) substrates: ZnCl_2_·6H_2_O, (PPh_4_)[W(CN)_6_(bpy)]. Time of crystal growth—8 months.

(b) substrates: ZnCl_2_·6H_2_O, (PPh_4_)[W(CN)_6_(bpy)], bpy. Yield 80%. Elemental analysis of 1·2.5H_2_O:C 39.20, N 17.83, H 2.80%. Calculated: C 39.12, N 17.55, H 2.65%

#### 4.2.2. Synthesis of [Cd(bpy)(H_2_O)(NO_3_)][W(CN)_6_(bpy)] (**2**)

(a) substrates: Cd(NO_3_)_2_·4H_2_O, (PPh_4_)[W(CN)_6_(bpy)]. Time of crystal growth—6 months.

(b) substrates: Cd(NO_3_)_2_·4H_2_O, (PPh_4_)[W(CN)_6_(bpy)], bpy. Yield 72%. Elemental analysis of 2·2Me_2_CO:C 39.73, N 16.09, H 2.79%. Calculated: C 40.00, N 16.03, H 3.15%

#### 4.2.3. Synthesis of [Cd(bpy)(H_2_O)][W(CN)_6_(bpy)]·2H_2_O (**3**)

(a) substrates: Cd(NO_3_)_2_·4H_2_O, (PPh_4_)_2_[W(CN)_6_(bpy)]. Time of crystal growth—4 months.

(b) substrates: Cd(NO_3_)_2_·4H_2_O, (PPh_4_)_2_[W(CN)_6_(bpy)], bpy. Yield 90%. Elemental analysis of 3·2Me_2_CO:C 41.52, N 16.07, H 2.99%. Calculated: C 41.43, N 16.66, H 2.88%

#### 4.2.4. Synthesis of {[Zn(bpy)(NO_3_)]}_2_[W(CN)_6_(bpy)]·2H_2_O (**4**)

(b) substrates: Zn(NO_3_)_2_·6H_2_O, (PPh_4_)_2_[W(CN)_6_(bpy)], bpy. Yield 95%. Elemental analysis of 4:C 39.20, N 17.83, H 2.80%. Calculated: C 39.33, N 17.84, H 2.57%

#### 4.2.5. Synthesis of Ferromagnetic Material **5** and **6**

The ferromagnetic material was prepared both for Zn^2+^ (**5**) and Cd^2+^ (**6**) cationic salts of both [W(CN)_6_(bpy)]^2−^ and [W(CN)_6_(bpy)]^−^ ions according to the methods described in patent applications, Refs. [11,12]. The PPh_4_^+^ salts of W(IV) (40 mg, 33 μM) and W(V) (116 mg, 132 μM) (1:4 molar ratio) and bpy (23 mg, 150 μM) were dissolved in 20 mL of the H_2_O-MeCN (1:1, *v/v*) mixture and heated to ca. 80 °C. Then, the saturated solution of Zn(NO_3_)·6H_2_O or Cd(NO_3_)_2_·4H_2_O (1 g, mM in 40 mL of water) was added and stirred intensively. The mixture was kept at ca. 90 °C for the next 10 min, cooled to room temperature, and the precipitate was filtered off, washed three times with water (3 × 20 mL), four times with acetone (4 × 5 mL) and dried in the air. Yield ca. 98%. The EDS map element distribution indicates the uniform distribution of W and Zn or Cd (see Appendix A). The EDS spectra of **5** and **6** shown in Appendix A show a Zn:W molar ratio of 1.24, confirming a 4:1 molar ratio of W(V)/W(IV) used in the synthesis. The Cd:W ratio found is 1.06, as both in **2** and **3**, the Cd:W ratio is 1:1.

### 4.3. Physicochemical Measurements

The elemental analyses of C, N and H were performed on the Elementar Vario MICRO Cube. The IR spectra were measured on the Nicolet iS5 FT-IT spectrometer in the 400–4000 cm^−1^ range in ATR mode. Magnetic susceptibility measurements were performed on a SHERWOOD SCIENTIFIC magnetic susceptibility balance. The thermogravimetric measurements were performed on a TGA/SDTA 851e Mettler Toledo Microthermogavimeter at a scan speed of 10 ^o^/min from 25 to 650 °C (to avoid sublimation of WO_3_) under argon. Compounds **5** and **6** were examined by scanning electron microscopy (SEM) Tescan VEGA 3 with a LaB_6_ emitter, equipped with the EDS detector (Oxford Instruments, X-act, SDD 10 mm^2^). The measurements were performed on a carbon sheet without gold sputtering. Dc magnetic measurements were performed with the use of a vibrating sample magnetometer (VSM) option of the Physical Property Measurement System (PPMS) produced by Quantum Design. The powder sample was placed in a sample container, which was mounted on a standard quartz sample holder. The sample container was fixed to the holder by the GE varnish. At the beginning, the sample was demagnetized at room temperature (300.0 K) by decreasing the oscillating magnetic field, and afterward, it was cooled down to 1.90 K at zero magnetic field (zero-field cooling (ZFC) regime). Next, the field of 1 kOe was switched on, and a measurement of the magnetic moment vs. temperature was taken within the temperature range from 1.90 up to 300.0 K (“ZFC, up” curve). Subsequently, the measurement at 1 kOe with temperature decreasing from 300.0 down to 1.90 K was run (“FC, down” curve), while the final measurement was performed again with the increasing temperature (“FC, up” curve). In order to collect the magnetic hysteresis loop, at the beginning, the sample was demagnetized at room temperature (300.0 K) by decreasing the oscillating magnetic field, and afterward, the desired temperature (i.e., 1.90 K or 300.0 K) was reached. Then, the isothermal measurement of the magnetic moment vs. applied magnetic field was performed using the following sequence: from 0 to 9 T, then from 9 to 9 T, subsequently from 9 to 9 T, and finally, from 9 to 0 T. With the aim of eliminating the diamagnetic contribution arising from the sample holder and the GE varnish, the same measurements as mentioned above were performed for a quartz sample holder with the same amount of GE varnish but without any sample mounted. The latter data were subtracted from the previous ones, leading to pure signal related only to the sample, which was then shown in Figure 4.

### 4.4. Single-Crystal Structure Determination

Single-crystal diffraction intensity data of compounds **1**, **2** and **3** were collected on a Rigaku XtaLAB Synergy-S diffractometer, mirror-monochromated at 100.0(1) K for **1** and Agilent Technologies SuperNova Dual with Atlas detector using mirror-monochromated Mo Kα radiation (λ = 0.71073 Å) for **2** and **3**, respectively, at 273.0(1) and 250.0(1) K.

Cell refinement and data reduction were performed using the CrysAlisPro firmware [16]. The positions of all non-hydrogen atoms were determined by direct methods using SHELXL-2017/1 [17,18]. All non-hydrogen atoms were refined anisotropically using weighted full-matrix least squares on *F*^2^. Refinement and further calculations were carried out using SHELXL-2017/1 [17,18]. All hydrogen atoms joined with carbon atoms were positioned with idealized geometries and refined using a riding model with *U*_iso_(H) fixed at 1.2 *U*_eq_ (C_arom_). The figures were produced using the Diamond ver. 4.6.1 software (Dr. H. Putz & Dr. K. Brandenburg GbR, Bonn, Germany) [19].

### 4.5. XRD Measurements of Powder Samples

Polycrystalline samples of compounds 5 and 6 were prepared using the method described previously. The XRD powder diffractograms of these materials (crystallinity tests) were recorded at 294 K on a Rigaku XtaLAB Synergy-S diffractometer with mirror image and monochrome mirror-monochromated CuK_α_ radiation using the default procedure included in the CrysAlisPro firmware [16].

## 5. Conclusions

We present the methods of synthesis of two high-temperature ferromagnetic materials based on cyanido complexes of W(IV/V). Due to different charge of cyanido anions, during synthesis, the W(IV) complexes start to crystalize, at first forming a template for further deposition of W(V) salts. As W(IV) complexes have a 2D polymeric structure, while those of W(V) are 1D polymers, this procedure results in the formation of spin interactions between the W(V) *d*^1^ centers responsible for ferromagnetic properties. These are stable up to the crystal decomposition (starting at ca. 150 °C); thus, the Curie temperature cannot be determined. The magnetic properties are, as expected, strongly dependent on the template [W(IV)] structure but also on the synthetic conditions, thus, the growth and symmetry of the W(V) complex. The use of two different cations (Cd^2+^ and Zn^2+^) shows that their size and properties influence the structures formed, thus, the ferromagnetic properties of products. The method presented, also checked for other cyanido systems, was patented by us and is promising as a starting point in future investigations on the enhancement of magnetic moment for similar systems.

## 6. Patents

Szklarzewicz, J.; Hodorowicz, M. Synthesizing a molecular magnetic material, US 17/067,777.

Szklarzewicz, J.; Hodorowicz, M. A molecular magnetic material and a method for preparation thereof, EP19183528.9.

## Data Availability

Not applicable.

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
