# Peer review of "The High-Temperature Soft Ferromagnetic Molecular Materials Based on [W(CN)6(bpy)]2−/− System"

_molecules, 2022, doi:10.3390/molecules27144525_

Round 1

Reviewer 1 Report

Dear Sir,
The paper is interesting and well prepared however it can be improved in some areas as follows:

The title

The title should be more focused on the main objective where it seems to be a title of  A review.

In the abstract

-          Please state exactly the performed analysis regarding the characterization such as XRD, etc.

-         keywords should be arranged alphabetically.

In the other sections:

-         Schematic diagram for the experimental part could be useful for the reader

-         All chemicals being used in the study should be addressed in this section including where they are purchased or obtained.

-         State the model, the company and country of origin of any instrument used in the research such as XRD.

-         EDX analyses could give further information on the chemical composition of the prepared samples

·        Future work should be mentioned at the end of the conclusion

References

Kindly, FORMAT the references correctly according to the author’s guide.

Regards

Reviewer 2 Report

‘’The high-temperature soft ferromagnetic molecular materials’’

by J. Szklarzewicz, M. Hodorowicz and A. Jurowska, ref. 1767401

This article focuses on the synthesis of molecular materials showing ferromagnetic properties.

First of all, the term ‘’soft’’ appearing in the title is not defined.

Second why ‘’high-temperature’’?

From a physical point of view, when inserting a diamagnetic ligand X between two magnetic cations A and B, a ‘’classical’’ insight leads to the fact the entity A-X-B may not show magnetic properties. This aspect was experimentally denied by Kramers at the beginning of the thirties (Physica 1, 182 (1934)). For explaining this phenomenon Kramers invented the term ‘’superexchange’’ but without giving a microscopic interpretation.

At the end of the fifties Anderson (Phys. Rev. 79, 350 (1950), Phys. Rev.115, 2 (1959), Solid State Phys. 14, 99 (1963)) as well as Kondo (J. Progr. Theor. Phys. 18, 541 (1957)) have shown separately that A-X-B generally shows antiferromagnetic properties because of superexchange: this phenomenon is due to the transfer of one electron of the ligand X characterized by a full s- or p-external shell into the d-external shell of the magnetic ion A (or B) for instance, through an excited state. In other words it has a pure quantum origin.

Unfortunately these theoretical models only worked for antiferromagnets but failed for describing ferromagnetic properties. However these models had the merit to propose a first quantum interpretation of superexchange.  As a result only empirical rules succeeded in giving the right sign of the exchange energy (i.e., the ferromagnetic or antiferromagnetic character of spin exchange between first-nearest neighbours). These rules proposed by Kanamori (J. Phys. Chem. Solids  10, 87 (1959)) and Goodenough (Phys. Rev. 100, 564 (1955), J. Phys. Chem. Solids 6, 287 (1958), Phys. Rev. 117, 1442 (1960)) perfectly worked for compounds A-X-B where the ligand X is simple i.e., with an average length of the same order as the magnetic cation one.

However, with the arrival of new magnetic polymers at the beginning of the nineties (A. Escuer,  R. Vicente, M. A. S. Goher, and F. A. Mautner, Inorg. Chem. 35, 6386 (1996); ibid, J. Chem. Soc. Dalton Trans., 22, (1997);  M. A. S. Goher, A. M. A-Y.Morsy, F. A. Mautner, R. Vicente, and A. Escuer, Eur. J. Inorg. Chem 2000, no 8, 1819 (2000); A. Escuer, J. Esteban, S. P. Perlepes,T. C. Stamatatos, Coord. Chem. Rev. 275, 87 (2014) etc…), the Goodenough-Kanamori rules failed due to the fact that ligand lengths were plainly much longer that the magnetic cation ones. Recently a full theoretical quantum model has been proposed by Curély (Magnetochem. 8, 6 (2022)) and gives the right sign of exchange energy as well as the true nature of spin-spin exchange (isotropic or anisotropic couplings).

All the possible strategies are recalled for establishing ferromagnetic couplings (see also Curély and Barbara, Struct. Bond., 122, 207 (2006)). I do not see in the paper one of these strategies. In addition there are no magnetic measurements so that there are no proofs that the described materials are ferromagnetic.

In addition, for 5d transition ions (as those of W) separated by ligands, we do not deal with linear entities (see Figs. 1 and 2): Jahn-Teller effect cannot be neglected, thus complicating the interpretation. This problem may be avoided if using 3d1 transition ions linked by ligands characterized by a long length. 

Finally, in conclusion, it is written that ‘’the size of cations Zn2+ and Cd2+ influences the structures formed (correct), thus the ferromagnetic properties…’’. The size of the cage in which ions Zn2+ and Cd2+ are inserted only plays a role at the level of the anisotropic character of exchange (classical coulombic problem). In addition it may or not influence the nature of spectrum of the entity A-X-B: generally strong coulombic interactions favor ferromagnetic couplings whereas weak ones favor antiferromagnetic couplings. In other words, the problem of ferromagnetism or antiferromagnetism has a pure quantum origin and is plainly more complicated than the explanations given in the article (see again Curély, Magnetochem. 8, 6 (2022)).

In summary this paper is not suitable for publication in Magnetochemistry.

Important changes must be brought: i) establishment of the crystallographic structure for each of the 1D, 2D and 3D compounds obtained with the unit cell [W(CN)6(bpy)]; ii) for each compound, thermal behavior of magnetization and susceptibility; dynamic and/or static hysteresis cycle thus proving that these compounds are soft ferromagnets.

Round 2

Reviewer 2 Report

Please see attached file suggesting revisions before a final acceptation.

Author Response

Thank you for additional comments on the manuscript. We have included a figure 4 and additional text fragments marked in green.